# Soft Tissue Reconstruction of the Posterior Trunk after Tumor Excision: A Surgical Algorithm

**DOI:** 10.3390/cancers15041214

**Published:** 2023-02-14

**Authors:** Marco Innocenti, Francesco Mori, Francesca Alice Pedrini, Luca Salmaso, Andrea Gennaro, Paolo Sassu

**Affiliations:** 1Dipartimento di Scienze Biomediche e Neuromotorie, Università di Bologna, 40126 Bologna, Italy; 2IRCCS—Istituto Ortopedico Rizzoli, 40136 Bologna, Italy

**Keywords:** soft tissue sarcomas, posterior trunk, perforator flaps, plastic surgery, reconstruction, flap

## Abstract

**Simple Summary:**

Reconstruction after the wide margin excision of a malignant tumor located in the posterior trunk has been considered challenging in the past. The paucity of conventional local donor sites and the concomitant increase in reconstructive requests, following progress in oncologic surgery, have combined to necessitate innovative solutions to cover even large defects by means of locoregional flaps. The advent of perforator flaps significantly increases the range of options available for reconstruction in this difficult area, thus minimizing donor site morbidity. The aim of this article is to summarize the current trends in the reconstruction of soft tissue defects in the posterior trunk.

**Abstract:**

Background: The posterior trunk has been considered a challenging area to reconstruct following soft tissue tumor excision because of the shortage of local donor sites. The advent of innovative procedures such as perforator flaps has radically changed this perspective and offered a new approach to the problem. Methods: Upon a review of the literature and the personal experiences of the senior author, an algorithm is developed according to the most updated procedure, combined with more conventional options that maintain a role in decision-making. Results: The upper back latissimus dorsi and trapezium flap are still the most reliable approaches, while perforator flaps based either on the circumflex scapular arteries or the transverse cervical artery represent a more refined option. In the middle third, few indications remain for the reverse latissimus dorsi, while the gold standard is represented by local perforator flaps based on the posterior intercostal system. In the lower back, conventional VY advancement flaps are still a safe and effective option in the sacral area, and perforator flaps based on posterior intercostal arteries, lumbar arteries and superior gluteal arteries are the first choice in most cases. Conclusions: Using perforator flaps significantly improved soft tissue reconstruction in the posterior trunk.

## 1. Introduction

Complex defects of the posterior trunk present challenges due to the paucity of conventional local options and shortages of recipient vessels for microsurgical free flaps. Causes of soft tissue defects include trauma, congenital malformation, spinal surgery and more frequently, oncological tumor resections.

Adult-type soft tissue sarcomas are rare; the most common are liposarcoma and leiomyosarcomas (LMSs), with an incidence of <1/100,000/year each [1,2]. In the posterior trunk, sarcomas and skin cancer can rapidly expand in both width and depth, often affecting the spine and eventually the spinal cord [3]. Furthermore, the spine is the most common site of bony metastasis, and the third most common site for solid-tumor metastasis [4,5].

The aim of the reconstruction is to ensure the adequate protection of the spinal cord, avoid leakage of the cerebrospinal fluid, and guarantee long-lasting coverage of the exposed bone and osteosyntheses hardware, thus preventing ulcers even in a prolonged supine position. Flaps are necessary in the irradiated surgical site because they carry the vascularized and viable soft tissue far from the fibrotic and sclerotic area.

In the recent past, common options have included the latissimus dorsi flap and trapezius in the upper back and reverse latissimus for the middle back, while free flaps have been considered almost the only option for the lower back.

The advent of perforator flaps has significantly improved the range of surgical solutions, as well as reduced the operative time and the morbidity at the donor sites [6]. Thus, the awareness of the high number of perforators in this anatomical region has increased the number of potential donor sites in areas next to the defect; it has also reduced the use of more invasive solutions.

In this article, we suggest an algorithm for soft tissue coverage of the posterior trunk in complex defects, covering conventional techniques to more innovative procedures.

## 2. Pertinent Anatomy

For reconstructive purposes, the upper third of the posterior trunk is primarily supplied by the thoracodorsal system, the transverse cervical artery and the dorsal and circumflex scapular arteries. Using these vessels, we can construct conventional muscular and myocutaneous flaps, such as the trapezium and latissimus dorsi flaps, or local perforator flaps supplied by perforators, such as the parascapular perforator flap [7,8,9,10,11].

The middle third cannot be entirely covered by the reverse latissimus dorsi, providing that the distal intercostal perforator to the muscle is present and healthy. Conversely, this region is rich in suitable perforators based on the posterior intercostal system, which can offer a range of choices to a surgeon in terms of flap elevation [6,7,12].

The lower third of the back is traditionally the domain of free flaps, although the only sizable recipient pedicle available in that area is the superior gluteal artery. VY flaps may be effectively used in sacral defects of small to medium sizes, but perforator flaps based on the lumbar arteries and the superior gluteal system nowadays offer a much wider range of options to cover even large defects [5,13,14,15].

## 3. Type of Flaps

Although many different classifications of flaps are offered depending on the area of consideration, the locoregional flaps that are currently used in covering the posterior trunk can be divided into three categories: advancement random flaps (keystone, VY), myocutaneous flaps and perforator flaps.

### 3.1. Advancement Random Flaps

These are skin flaps that are designed according to geometrical rules; they are formed from the large amount of tissue still connected to the fascia, and are well-vascularized by a perforator randomly located in the area of the flap. Larger flaps are preferred as they can be more widely supplied by a larger number of perforators. The incision must include the fascia of the underlying muscles, and the whole perimeter of the flap must be free. Undermining below the fascia may further increase the advancement of the flap, which seldom exceeds 4–5 cm. For this reason, particularly in midline defects, a second keystone flap mirroring the contralateral one is simultaneously harvested, making possible the coverage of wider longitudinal defects (Figure 1). Similar approaches can be applied in VY advancement flaps, which are particularly indicated for use in the lumbosacral region.

### 3.2. Myocutaneous Flaps

These comprise the latissimus dorsi and the trapezius (Figure 2). Both the muscles and the overlying skin are expendable and provide robust multilayer coverage that is particularly useful in cases of cerebrospinal liquid leakage. However, some functional and aesthetical morbidity occurs at the donor sites. In some circumstances, the same flaps can be elevated as pure muscle flaps, thus reducing the morbidity at the donor site, but also providing less resistant coverage that requires a skin graft.

### 3.3. Locoregional Perforator Flaps

The concept of the angiosome was introduced by Taylor [16] in 1987. It has revolutionized soft tissue coverage in all areas of the body, in that tiny perforators supply the subdermal plexus of the skin and can be used as the only pedicle of skin flaps [17]. Each perforator in the body is responsible for the supply of blood to a well-defined portion of skin called an angiosome (Figure 3). The so-called “anatomical” territory of any single perforator can be significantly expanded to adjacent angiosomes via direct and indirect linking vessels [18], which supply the “dynamic” and “potential” territories. Conversely, traditional fasciocutaneous “axial” flaps are based on a major vascular axis, which must be included in the flap, thus limiting the potential donor sites and representing a more invasive approach because they sacrifice a major vascular bundle [19]. Instead, the perforator flaps can be virtually elevated anywhere a sizable perforator is available [20], and can provide surprisingly large flaps in the area neighboring the defect without the sacrifice of important vessels, thus achieving the goal of reconstructing in a “like with like” fashion, which is one of the paradigms of plastic surgery. The local perforator flaps can be raised either as transposition flaps or, more frequently, as propeller flaps. In the transposition flap, once the perforator has been visualized and dissected, the skin is advanced to its recipient area, which must be nearby. Propeller flaps are usually larger elliptical flaps, with the perforator located in the proximity of the proximal pole [20] (Figure 4). The perforator must be dissected by at least 2 cm; it acts as the pivot of the flap, which can be rotated up to 180° in order to reach the recipient area, which may thus be relatively far from the donor site [21,22,23]. Propeller flaps are more versatile—they can reach more distant defects; therefore, they are preferred over transposition flaps.

## 4. Reconstruction by Districts

### 4.1. Upper Back

Defects on the midline are relatively common due to the tension in sutures or to radiotherapy. The gold standard for use in the cervical region is probably the trapezius flap, which is myocutaneous or a perforator based on the superficial branch of the transverse cervical artery (Figure 5). The latter is less invasive, but also more difficult to dissect [24]. More distally, at the cervicodorsal level, longitudinal defects on the midline less than 5 cm in width may be optimally repaired with 2 conventional key stone flaps. Alternatively, propeller flaps may be raised on the perforator of the circumflex scapular artery, particularly in the case of defects located on the scapular area (Figure 6). Additionally, the myocutaneous latissimus dorsi may be used in cases of wider defects next to the midline.

### 4.2. Midback

The propeller flaps based on branches of the posterior intercostal arteries currently represent the first choice for covering defects in this area (Figure 7).

The only conventional options are the reverse latissimus dorsi and keystone flaps. The reverse LD is based on the distal intercostal arteries, which are the secondary pedicle of the muscle [25,26,27]. However, the arch of rotation is limited, and the distal pedicle is not always reliable and healthy. Keystone flaps are indicated only in cases of relatively small longitudinal defects near to the midline.

### 4.3. Lower Back

Traditionally considered a no man’s land for local flaps, this region has been mainly covered by free flaps in the past [28]. However, the paucity of recipient vessels suitable for anastomosis has made this option challenging, often requiring vein grafts, with the superior gluteal artery being the only available recipient pedicle in this area. Once again, local perforator flaps significantly increase the potential area of local skin that can be used for reconstruction. At this level, using the perforator from the posterior intercostal arteries is an option, but perforators from the lumbar arteries or from the superior gluteal arteries are more often used. On the latter pedicle, huge flaps can be raised—bilaterally, if needed—thus providing enough skin to cover large defects (Figure 8). For small/medium defects of the skin overlying the sacrum, a VY advancement flap, either unilateral or bilateral, is a simple and effective option that should be considered as the first choice (Figure 9).

## 5. Discussion

Many new techniques have been developed in the area of soft tissue reconstruction in the posterior trunk—a particularly challenging region for traditional procedures. In 2006, Mathes et al. [10] concluded that the majority of defects in the back should be treated by muscle flaps; Hallock, in 2011 [7], suggested using myocutaneous flaps in the upper back and perforator flaps in the mid and lower back; and finally, Behr et al., in 2014 [29], championed using local perforator flaps for the entire posterior trunk.

In our opinion, the reconstructive surgeon should master several techniques, either traditional or innovative, and apply them in a customized manner to every single patient. In the oncological field, many variables should be considered in order to choose the best option. The extension of the involved skin and the neoplastic contamination of the deep planes are the first variables to be considered. In cases of full-thickness defects involving the vertebral spinous process, a myocutaneous flap is probably the best option to fill the dead space and provide more stable coverage. This is particularly true in cases of leakages of cerebrospinal fluid, which can be better controlled by means of a muscle, making this the best option for covering the irregular contours of the wound. An alternative technique may consist of the advancement of the paravertebral muscles underlying the skin defect—in order to cover the spine—and then using a perforator flap to reconstruct the skin in a tension-free manner.

Paraspinous muscles are common sources of muscle flaps for reconstructing the deepest portion of midline wounds in the posterior trunk, as these flaps take up all the space and provide a rich vascular supply that promotes healing [5]. In our practice, we advance the use of the muscle as a bipedicle muscle flap based on the lateral row of the intercostal artery. In some cases, paraspinous muscles may be removed due to oncological excision; hence, we advance using the trapezius muscle or the latissimus dorsi in the midline to cover the spine and the hardware, and to obliterate dead space.

Many patients have received heavy radiation therapy, which, in the long term, damages the skin neighboring the former surgical incision and necessitates late reconstruction. This is a common reason for the late dehiscence of surgical wounds, usually followed by exposure of the hardware. In this case, a wide resection of the irradiated skin is mandatory to ensure good results, and the flap must be suitably large. Usually, the defect is limited to the skin, and perforator flaps are the first choice; however, in cases of exposed metal devices, the paravertebral muscles should be bilaterally mobilized and sutured on the midline in order to provide better coverage and reduce the tension on the skin (Figure 10). In all cases, a wide excision of unhealthy skin up to well-vascularized margins, as well as tension-free skin sutures, are the prerequisites for a successful surgery.

Geddes et al. [30] described an average of 24 sizeable perforators in the upper back, ranging from the thyrocervical trunk to the subscapular system and the posterior intercostal arteries. More distally, several perforators are offered by the posterior intercostal arteries and the lumbar arteries. Additionally, the superior gluteal artery provides reliable perforators that are able to supply a large area of skin that can be rotated to the lower back in order to cover large defects, particularly if raised bilaterally [30].

Given the high number of perforators available in the posterior trunk, pedicled perforator flaps are nowadays the gold standard when dealing with most of the soft tissue defects in this anatomical area. They offer the advantage of reconstructing, in a “like with like” manner, even large defects by using multiple propeller flaps derived from different adjacent areas, with surprisingly low morbidity at the donor sites (Figure 11). These may be considered “microsurgical, nonmicrovascular” procedures, which require an experienced microsurgeon. Although they do not require microvascular anastomosis, the small size of the perforators (ranging 0.3–0.7 mm) requires the careful dissection of the pedicle under optical magnification with microsurgical instruments. These can then be advanced either as a transposition flap or as a propeller flap, the latter being more versatile and able to cover more distant defects, although also being more prone to complications, which may occur more frequently. More often, this involves the epidermolysis and superficial necrosis of the skin, but partial and total loss of the flap may also be included in the possible complications.

Random advancement flaps, such as VY and K stone flaps, represent a simple and reliable option, particularly for use in the sacral and dorsal region for small/midsize defects.

We summarize our surgical strategy for the soft tissue reconstruction of the posterior trunk in our flowchart (Figure 12). We propose an algorithm based on the presence or absence of bone/hardware exposure and the site of the defect. In cases of bone or instrumentation exposure, we prefer the use of a stronger cover with muscle flaps or thick perforating flaps. Specifically, in the upper back, we use the latissimus dorsi flap or trapezius flap, while for middle-back defects we harvest reverse latissimus dorsi flaps or advance the paravertebral muscle to the midline, which we cover with a perforator or keystone flap; in the case of lower-back defects, we use superior gluteal artery perforator flaps or lumbar artery perforator flaps. When there is no bone or instrumentation exposure, we prefer not to use muscle flaps in order to reduce functional damage; we usually prefer perforator flaps. For the upper back, we prefer to use a trapezius perforator flap or parascapular perforator flap, while in the middle back we use keystone flaps, trapezius perforator flaps or posterior intercostal perforator flaps. For the lower back, we tend to harvest a superior gluteal artery perforator flap, a lumbar artery perforator flap or a VY advancement flap. Due to the scarcity of recipient vessels in the posterior trunk and the surgical commitment for the patient, we usually prefer not to resort to free flaps.

## 6. Conclusions

Posterior trunk soft tissue reconstruction benefits from the new perforator-based reconstructive paradigm, which significantly increases the number of donor sites adjacent to the defect. Hence, this provides large amounts of vascularized skin with low morbidity at the donor site, which can be closed in the majority of cases. In select cases, using conventional advancement flaps is still a reasonable option for a small/medium defect in certain areas of the back. Free flaps are to be used only in special cases where other options could not be adopted.

## Figures and Tables

**Figure 1 cancers-15-01214-f001:**
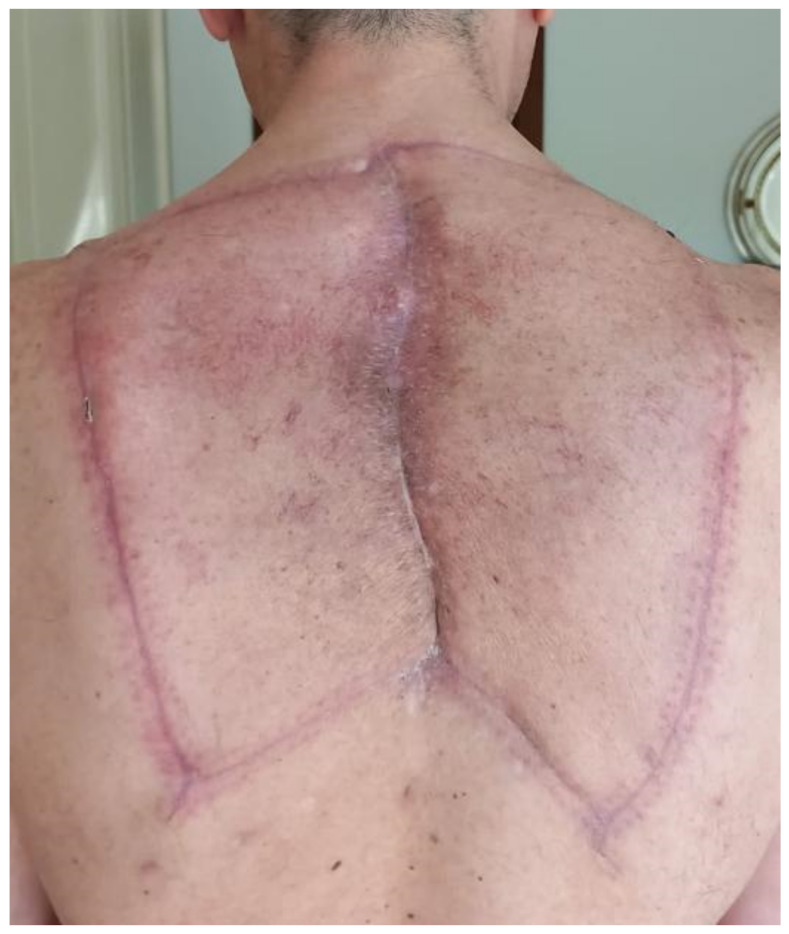
Follow-up control of double keystone flap in wound dehiscence in irradiated skin. The defect on the midline is longitudinal and relatively small in width; it can be easily covered by two key stone flaps.

**Figure 2 cancers-15-01214-f002:**
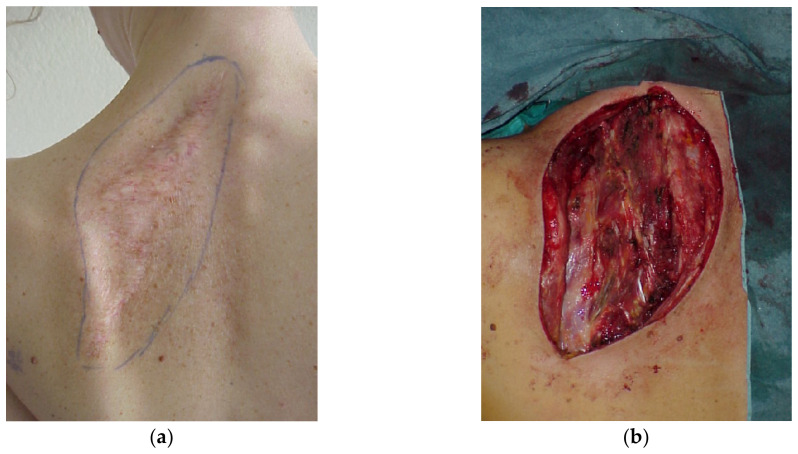
(**a**) Recurrence of fibrosarcoma in the scapular region; (**b**) the oncological excision exposes the scapula, (**c**) the tumor has been excised and (**d**,**e**) a pedicled myocutaneous latissimus dorsi flap has been used to cover the full thickness of the defect.

**Figure 3 cancers-15-01214-f003:**
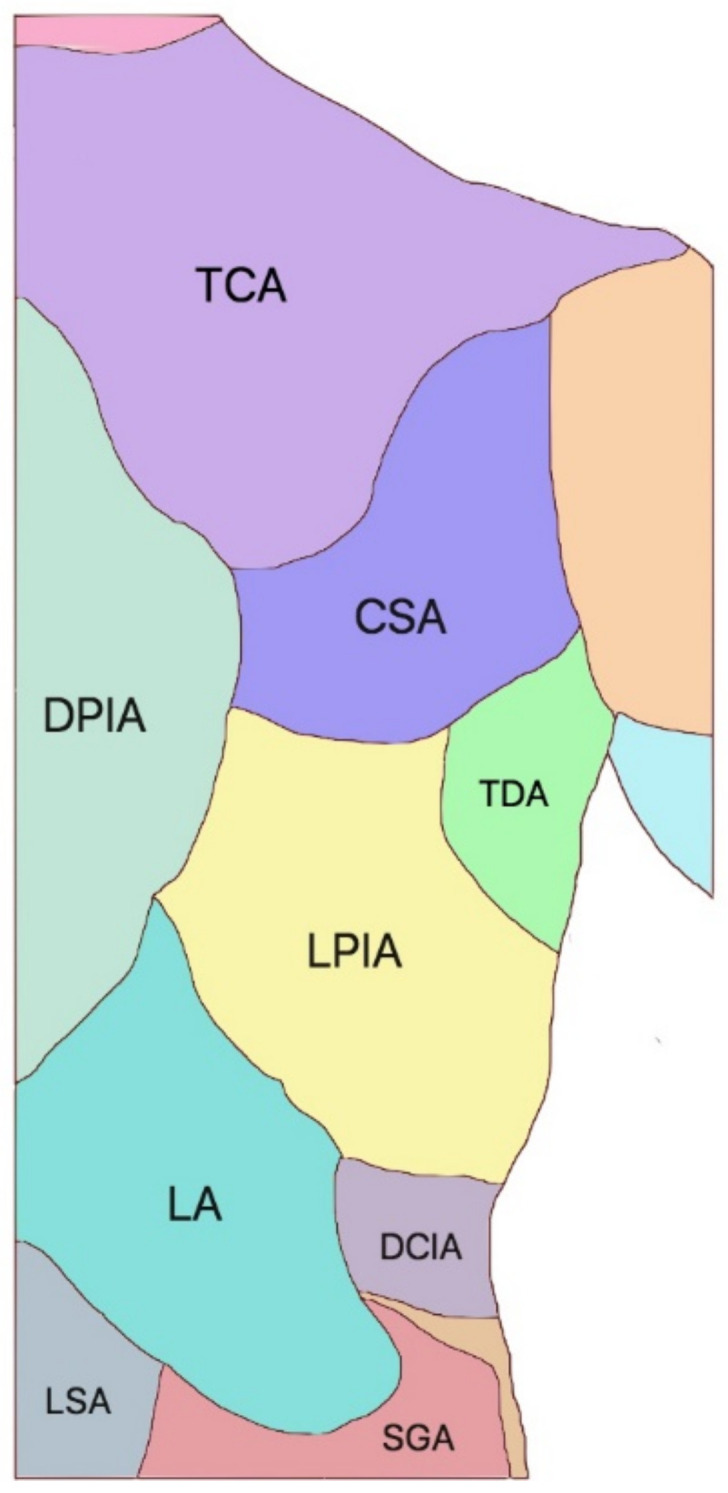
Right half posterior trunk and vascular territories. TCA, transverse cervical artery; DPIA, dorsal branch of the posterior intercostal artery; CSA, circumflex scapular artery; TDA, thoracodorsal artery; LPIA, lateral branch of the posterior intercostal artery; LA, lumbar artery; DCIA, deep circumflex iliac artery; SGA, superior gluteal artery; LSA, lumbosacral artery.

**Figure 4 cancers-15-01214-f004:**
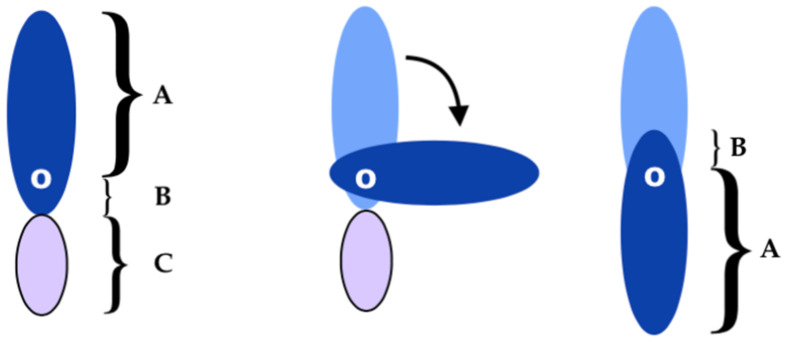
Propeller flap. On the left side, the planning of the elliptical flap. A + B is the flap; O is the pivot point (where the perforator is located) and C is the injury zone receiving the flap.

**Figure 5 cancers-15-01214-f005:**
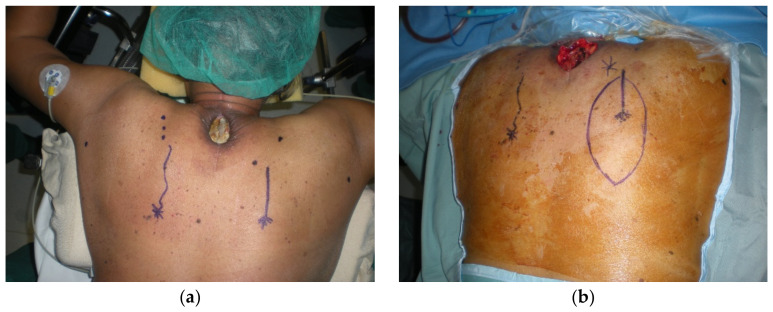
Wound dehiscence after radiotherapy at the cervicodorsal level. Coverage with a perforator flap based on the superficial branch of the transverse cervical artery. (**a**) Preoperative image of the defect; (**b**) design of the flap; (**c**) harvested flap; and (**d**) 90 days post-operative follow-up.

**Figure 6 cancers-15-01214-f006:**
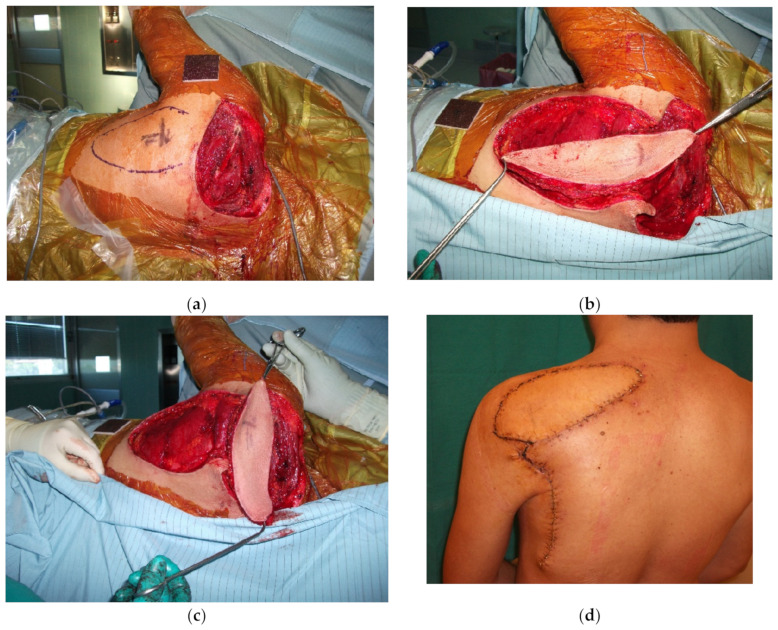
Raising of the pedicle from the circumflex scapular artery has been used in the past to supply the scapular and the parascapular free flaps. On the same perforator, a propeller flap can be elevated to cover more proximal defects, such as in the case shown here after the excision of a dermatofibrosarcoma. (**a**) The residual defect in the scapular region; (**b**) the flap elevated on the perforator of the circumflex scapular artery; (**c**) 90° rotation of the flap; and (**d**) two-week post-operative follow-up. The donor site is closed directly.

**Figure 7 cancers-15-01214-f007:**
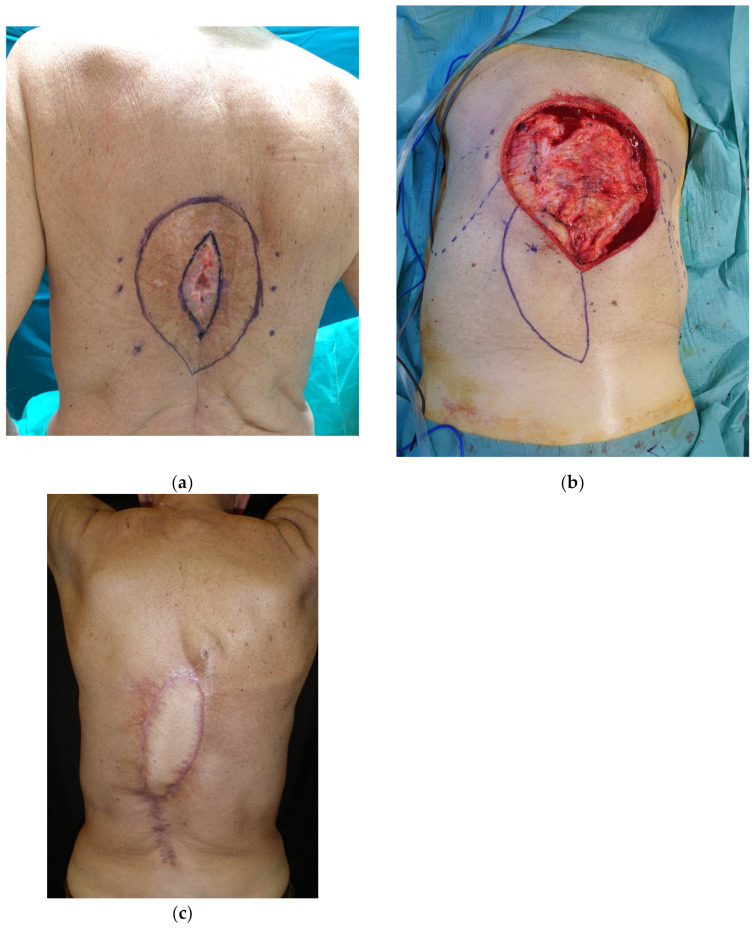
(**a**) Dehiscence after attempt at direct closure in irradiated skin; (**b**) wide margin excision of the compromised skin and design of a propeller flap based on the perforator of posterior intercostal artery; and (**c**) result at one month post-operation.

**Figure 8 cancers-15-01214-f008:**
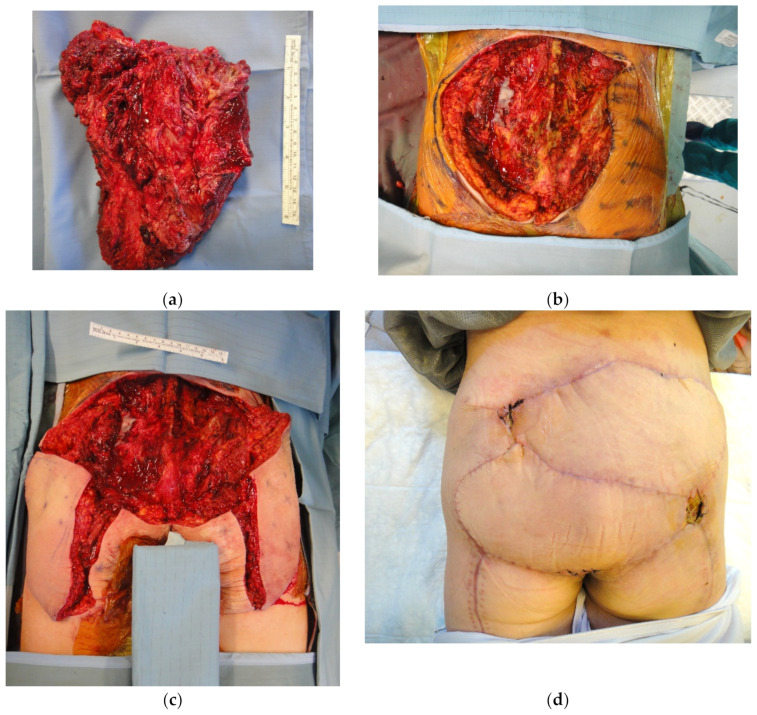
(**a**) Specimen of fibrosarcoma after wide margin excision; (**b**) the residual defect—22 × 16 cm; (**c**) 2 propeller flaps elevated on the perforator of the superior gluteal artery bilaterally; and (**d**) result at 2 months FU (Follow-up).

**Figure 9 cancers-15-01214-f009:**
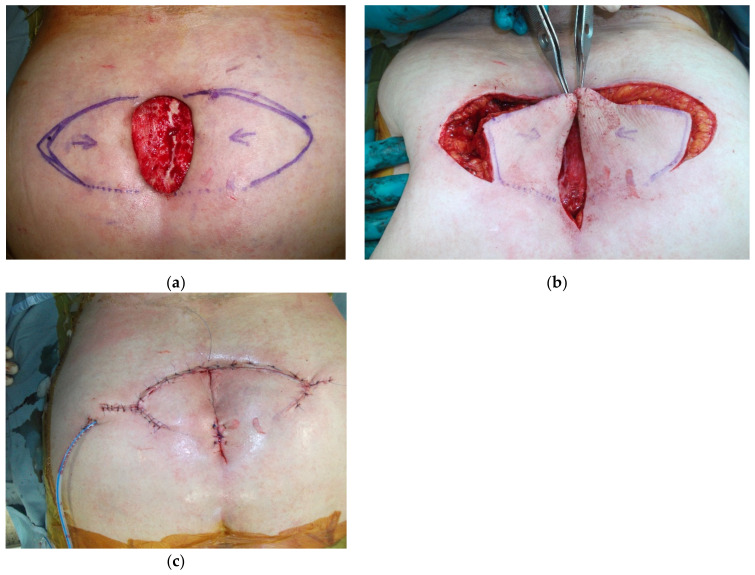
(**a**) Excision of squamous cells carcinoma in the sacral region; (**b**) two elevated VY flaps; and (**c**) result at the end of surgery.

**Figure 10 cancers-15-01214-f010:**
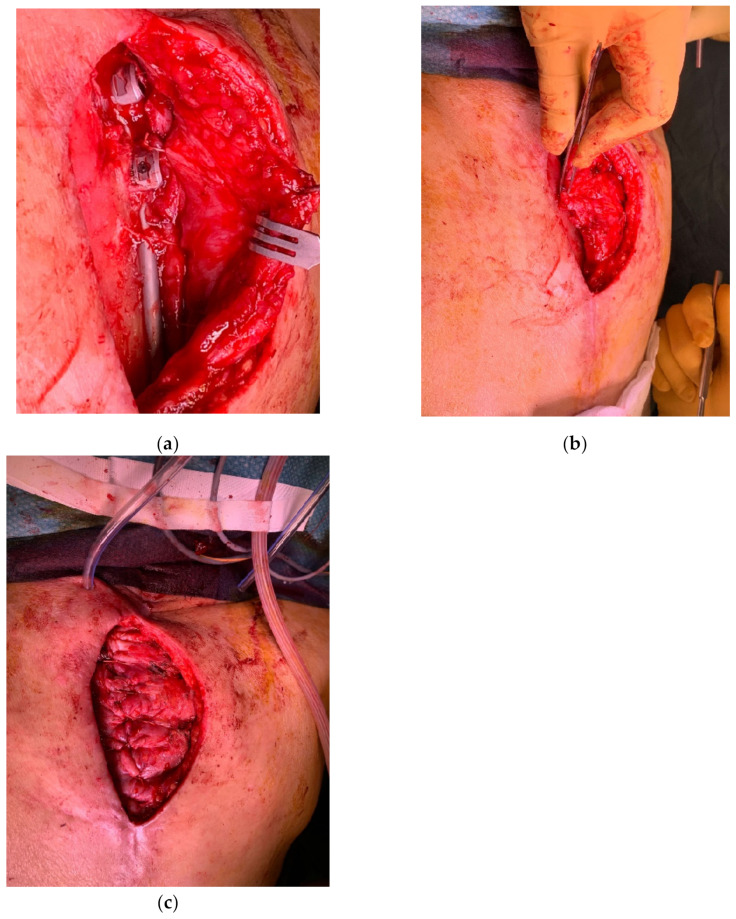
(**a**) Exposure of titanium bars in the dorsal spine; (**b**) the paravertebral muscles mobilized on both sides; and (**c**) tensionless suture and coverage of the hardware.

**Figure 11 cancers-15-01214-f011:**
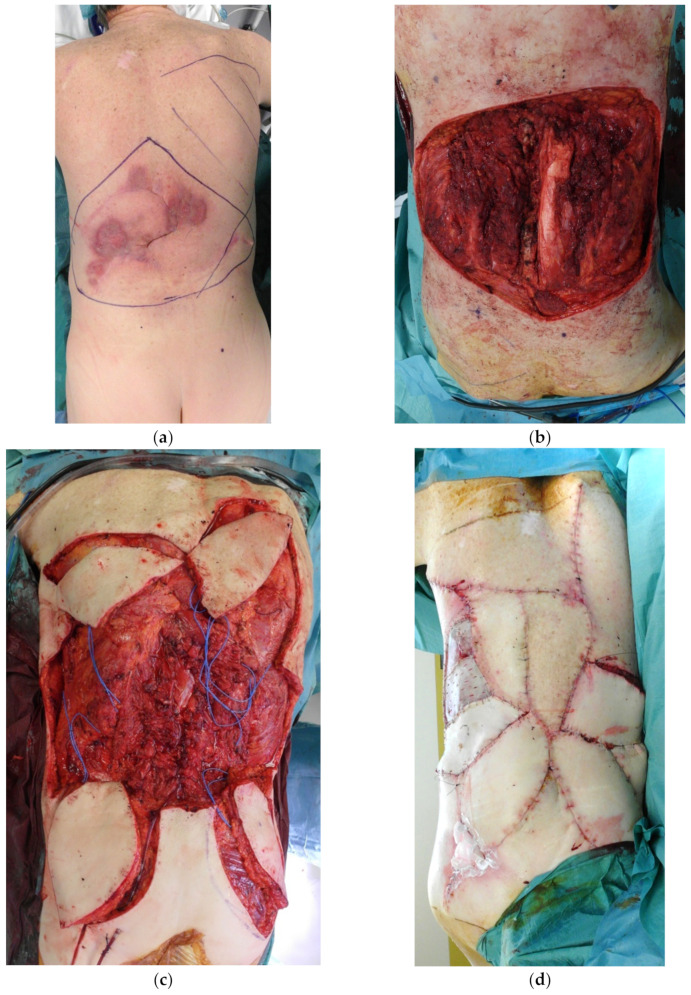
(**a**) Recurrence of squamous cell carcinoma infiltrating the deep planes; (**b**) a soft tissue defect (28 × 19 cm) with exposure of the spinal cord resulting from the tumor excision; (**c**) 6 propeller flaps were raised on perforators from the posterior intercostal and lumbar networks; (**d**) skin grafting was needed in only 1 donor site; and (**e**) the result at 4 months FU.

**Figure 12 cancers-15-01214-f012:**
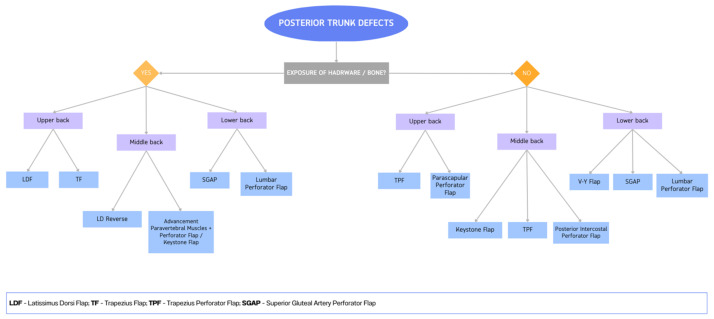
Flowchart for posterior trunk defects reconstruction.

## Data Availability

For all data requests, please contact the corresponding authors.

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
