# Peer review of "Soft Tissue Reconstruction of the Posterior Trunk after Tumor Excision: A Surgical Algorithm"

_cancers, 2023, doi:10.3390/cancers15041214_

Round 1
Reviewer 1 Report
This review aims to provide updated guidelines for post-oncological soft tissue reconstruction of the posterior trunk using locoregional flaps. The locoregional flaps currently used in the coverage of the posterior trunk belongs to three categories: advancement random flaps (keystone, VY), muscle or musculocutaneous flaps and perforator flaps; instead of free flaps which should be reserved to cases where no other options could be adopted. The employment of locoregional pedicled flaps is described and discussed for reconstruction of posterior trunk’s districts (upper back, midback, and lower back) according to the size of defect.
The Abstract and the Maintext are well-written. The authors provide a good overview of the reconstruction of the posterior trunk, currently lacking in the Literature, and traditional or perforator flaps available to restore different districts. The text could be implemented with a schematic algorithm of the possible reconstructive options for the posterior trunk.
Author Response
Kind reviewer, thank you for your review and comments.
We have added the flowchart (Figure number 12) as a representative diagram of the algorithm you asked for.
Reviewer 2 Report
Review:
Soft tissue reconstruction of the posterior trunk after tumor ex-2cision: a surgical algorithm
I read with interest the paper describing the authors approach to the treatment of surgical defects on the posterior trunk. The authors explain their rational and their choice of flaps as well their departure from the “classic” muscle flaps to a preferred use of perforator fasciocutanous flaps. This should be presented as an algorithm – it is not a guidline as guidelines are usually made by organization using a consensus process.
The message is understood but is not clearly presented there are many English mistakes that make it difficult to follow and sometimes convey the wromg message.
To name a few:
Miocutaneous should be myocutanous,
Trapezium should be trapezius
Mayor should be major
“flap result to be still the conventional workhorses” This is not an English sentence
“On those vessels can be based conventional muscular and miocutaneous flaps”
“The perforator flaps, instead, can be virtually elevated anywhere a sizable perforator is available”
The introduction is long on the treatment of sarcomas, in my opinion it should focus on the reconstructive challenges which are: radiated tissue, exposed bone, exposed osteosyntheses hardware, possible cerebrospinal leakage and the need to bear weight when prone and avoid ulceration on pressure points.
Page 2 row 82: I disagree, V-Y flaps can advance more, this is also dependent on the location and the thickness of the subcutaneous tissue
Author Response
Dear Reviewer thank you for your correction and comments.
Please find attached our corrections.
1- "This should be presented as an algorithm – it is not a guidline as guidelines are usually made by organization using a consensus process."
1 - We agree with you and we corrected the concept by removing the term guidelines
2- "The message is understood but is not clearly presented there are many English mistakes that make it difficult to follow and sometimes convey the wrong message."
2- We sent the manuscript to the journal's specialized english reviewers to correct all English mistakes.
3- "The introduction is long on the treatment of sarcomas, in my opinion it should focus on the reconstructive challenges which are: radiated tissue, exposed bone, exposed osteosyntheses hardware, possible cerebrospinal leakage and the need to bear weight when prone and avoid ulceration on pressure points."
3- Thank you for your comment, we have edited the introduction following your suggestions.
4- "Page 2 row 82: I disagree, V-Y flaps can advance more, this is also dependent on the location and the thickness of the subcutaneous tissue"
4- Thank you again for your comment, we have edited the part you suggested to correct.